# Dimensional reduction of emergent spatiotemporal cortical dynamics via a maximum entropy moment closure

**Yuxiu Shao**[1], **Jiwei Zhang**[2], **Louis Tao**[1,3]*

**1** Center for Bioinformatics, National Laboratory of Protein Engineering and Plant Genetic Engineering, School of Life Sciences, Peking University, Beijing, China, **2** School of Mathematics and Statistics, and Hubei Key Laboratory of Computational Science, Wuhan University, China, **3** Center for Quantitative Biology, Peking University, Beijing, China

\* taolt@mail.cbi.pku.edu.cn

**Data Availability Statement:** All relevant data are within the manuscript and its Supporting Information files.

**Funding:** Acknowledgments. This work was partially supported by the Natural Science

## Abstract

Modern electrophysiological recordings and optical imaging techniques have revealed a diverse spectrum of spatiotemporal neural activities underlying fundamental cognitive processing. Oscillations, traveling waves and other complex population dynamical patterns are often concomitant with sensory processing, information transfer, decision making and memory consolidation. While neural population models such as neural mass, population density and kinetic theoretical models have been used to capture a wide range of the experimentally observed dynamics, a full account of how the multi-scale dynamics emerges from the detailed biophysical properties of individual neurons and the network architecture remains elusive. Here we apply a recently developed coarse-graining framework for reduced-dimensional descriptions of neuronal networks to model visual cortical dynamics. We show that, without introducing any new parameters, how a sequence of models culminating in an augmented system of spatially-coupled ODEs can effectively model a wide range of the observed cortical dynamics, ranging from visual stimulus orientation dynamics to traveling waves induced by visual illusory stimuli. In addition to an efficient simulation method, this framework also offers an analytic approach to studying large-scale network dynamics. As such, the dimensional reduction naturally leads to mesoscopic variables that capture the interplay between neuronal population stochasticity and network architecture that we believe to underlie many emergent cortical phenomena.

## Author summary

Emergent nonlinear dynamics in the primary visual cortex (V1) may influence information processing in the early visual pathway and has been shown to affect visual perception. A major goal of systems neuroscience is to understand how complex brain functions can arise from the collective nonlinear dynamics of the underlying neuronal network. This challenge has been partly met through electrophysiological recordings, optical imaging and neural population models. However, a full account of how the multi-scale population

Foundation of China through grants 11771035 (J. Z.), 31771147 (Y.S., L.T.) and 91232715 (L.T.) and by the Open Research Fund of the State Key Laboratory of Cognitive Neuroscience and Learning grant CNLZD1404 (Y.S., L.T.). The funders had no role in study design, data collection and analysis, decision to publish, or preparation of the manuscript.

**Competing interests:** The authors have declared that no competing interests exist.

dynamics emerges from the detailed biophysical properties of individual neurons and the network architecture remains elusive. Previously, working on a homogeneously-coupled network, we derived a series of population dynamics models, ranging from Master equations, to Fokker-Planck equations, and culminating in an augmented system of spatially-coupled ODEs. Here we present an application of this reduction method to a heterogeneously coupled neuronal network that models a spatially-extended portion of V1. We found that the temporal dynamics of individual V1 patches can be well captured by a low-dimensional set of voltage moments. At the same time, the spatially-coupled system can recapitulate the cortical wave generation and propagation induced by many visual stimuli, including those that induce motion illusions. Furthermore, this coarse-graining reveals the importance of the temporal differences between on-/off-pathways, that may account for the directional motion perception from darks to brights.

## Introduction

Multi-channel recordings and optical imaging have revealed a vast repertoire of spatiotemporal activity patterns in the brain. This rich hierarchy ranges from localized activation to traveling waves, to dynamically switching cortical states [1–4]. The activities can be stimulus-driven or internally generated and are shown to affect information processing, sensory perception and cognitive tasks [5–11]. Mathematically, the emergence of the many spatial and temporal scales in cortical dynamics presents a tremendous challenge for modelers and theoreticians. The rapid development in computational power has allowed us to study very large networks, and a combination of large-scale network simulations [12–15], reduced dimensional models (e.g., neural field models, mean-field populations models, and kinetic theories [3, 6, 9, 16–19]), and machine learning [20–22] have been used to successfully describe many experimental phenomena. The principal mechanism underlying the diverse spectrum of network activities is likely to be the strong competition between excitatory and inhibitory neuronal populations. However, a theoretical account of how the detailed biophysical properties of individual neurons, the local network properties and cortical architecture can lead to the observed emergent multi-scale dynamics is lacking. Here we make progress towards such a theoretical model by making use of a coarse-graining formalism that has been successful at capturing the rich repertoire of heterogeneous dynamics that exists even in a small homogeneous neuronal network. Recently, working on a small, idealized network of linear integrate-and-fire neurons, through using a partitioned-ensemble-average (PEA), we derived a sequence of population dynamics models, ranging from Master equations, to Fokker-Planck models, and finally to an augmented system of ODEs that explicitly accounts for the interaction between neuronal spiking activity and internal neuronal voltages [23].

In [23] (hereafter ZSRT19), from the Fokker-Planck description of neuronal population dynamics, we derived an augmented low-dimensional ODE system by introducing a hierarchy of neuronal voltage moments and a maximum entropy closure. We showed that, by carefully introducing the PEA into our simulation algorithms, our dimensional reduced population models could faithfully capture highly heterogeneous network dynamics, ranging from transient to sustained, from driven to self-organized, from oscillatory to nearly synchronous network activity in the form of multiple-firing events. More importantly, the PEA formalism provided a conceptual framework to mathematically coarse-grain the emergent network dynamics from first principles. However, so far, the applications have been restricted to small networks with homogeneous connectivities, which is an idealization that is met in but the

smallest of local cortical networks. Here we extend our methodology to networks with slowly varying spatial inhomogeneities and apply it to study the emergent dynamics of the primary visual cortex (V1).

The mammalian V1 is of particular interest to many neuroscientists owing to its fundamental role in visual processing and the common belief that understanding network functions in V1 will advance our understanding of other areas of the mammalian brain. While individual V1 neurons show preference to orientation of a visual stimulus by elevated firing rates, optical imaging experiments show that orientation preference is distributed in pinwheel-like hypercolumns that tile the cortical surface. Recent optical imaging techniques, particularly, voltage-sensitive dye (VSD) imaging, can capture V1 network dynamics with high spatial and temporal resolution and revealed the important interplay between visual stimulus, subthreshold population dynamics and large-scale, coherent activities [1, 2, 24–26].

The application of the ZSRT19 formalism to V1 naturally produces a spatially-coupled ODE system, consistent with locally organized visual feature maps. By examining local network patches smaller than orientation hypercolumns, we show that sub-hypercolumn temporal dynamics can be well captured by a low-dimensional set of voltage moments. By adding orientation specific couplings between orientation hypercolumns, we were able to recapitulate the cortical wave generation and propagation induced by visual illusory stimuli. Finally, by modeling the temporal difference between On- and Off-visual pathways as temporal differences of the respective inputs into V1, our reduced-dimensional model can account for the induction of propagating voltage waves from darks to brights.

## Results

The effectiveness and accuracy of our coarse-graining framework has been well investigated in various aspects, see [23, 27]. Here we begin with validating our moment closure by applying it to model the network response to a rotating drifting-grating stimulus. Here, the visual stimulus is a sinusoidally modulated grating, drifting at 4 Hz and rotating at 20˚/sec. We numerically simulate one single orientation hypercolumn (see Fig 1A and Methods). Fig 2 shows, over one rotation period, the temporal dynamics of the total synaptic inputs (Fig 2A), which consist of the external (i.e., LGN) input (Fig 2B), excitatory cortical input (Fig 2C) and inhibitory cortical input (Fig 2D). (The faster oscillations correspond to the drift rate of the sinusoidally modulated grating.) Each panel plots the dynamics at two locations with respect to the pinwheel center, with red/blue representing neuronal populations far from/near the pinwheel center, respectively. Clearly, the neurons away from the pinwheel center (in the so-called iso-orientation domain) have flatter temporal responses compared to neurons near the pinwheel center. Since the grating is also slowly rotating, the envelope of the temporal response curve can be used to estimate the orientation tuning of each neuronal population. Fig 2E displays the population-averaged membrane potentials and slaving-voltages (see Methods) of individual patches (marked in Fig 2F–2H); red/blue solid lines are population-averaged membrane potentials of neurons away/near the pinwheel center; pink/cyan dash lines are corresponding slaving-voltages. Fig 2F–2H are spatial patterns of population-averaged membrane potential and population firing rate at $t$ = 3475 $ms$. From Fig 2E–2H we can infer that the neurons near the pinwheel center are more selective to stimulus orientation while the neurons far from the pinwheel center are less selective, consistent with the network models of McLaughlin & Rangan [28].

Population activity in cortex forms characteristic clusters both in space and time. While the processing of local, small-scale stimulus orientation appears to be performed within individual orientation hypercolumns, the integration and processing of more global features are believed

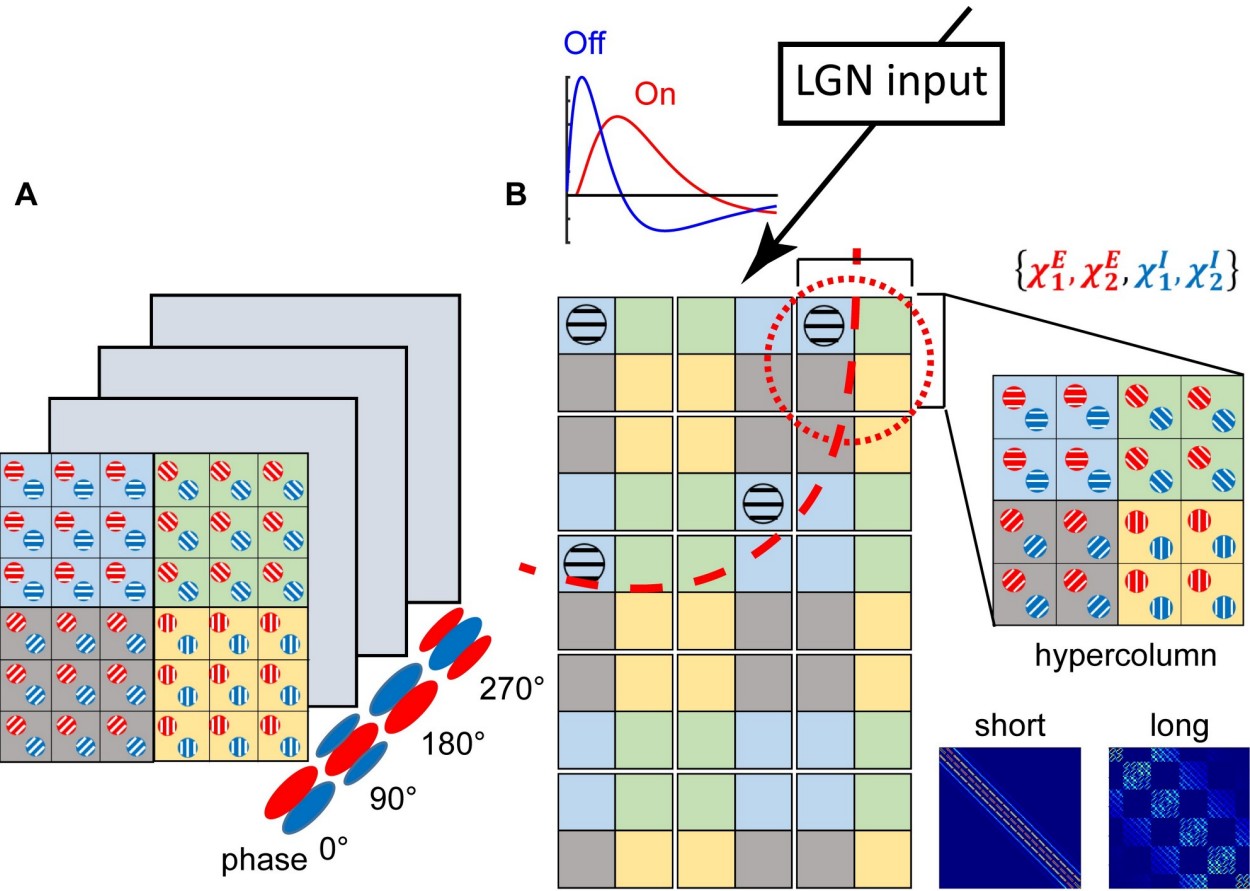

**Fig 1. Schematic diagrams of network architectures used in simulations.** (A) shows a single-hypercolum structure used in the first simulation. Neuronal populations in these four layers are nearly identical, except for their preferred phases. (B) shows the second network structure in the simulation. The reduced model contains 5×3 hypercolumns (red, dashed circle), each of which has 4×4 CG patches. The patch contains both the excitatory (red) and inhibitory (blue) subpopulations, thus can be described by 4 voltage moments (upper right). These patches are colored according to their preferred orientations (PO) and connected via short-range as well as long-range couplings. The short-range couplings are isotropic and constrained within a single hypercolumn (the 1st part in the bottom right panel), while the long-range couplings are NMDA-type, excitatory and orientation-dependent (red, dashed arc in the left panel and the 2nd part in the bottom right panel). LGN input and corresponding temporal profiles are shown in the top. See Methods for more details.

to be functions of a network organization on scales spanning multiple hypercolumns. Furthermore, many experimental studies have revealed the existence of spatially separated On- and Off-visual pathways [29–31], in addition to temporal differences, originating from the RGCs and persisting to V1. On the scale of single neurons, the well-known spatial arrangement of LGN inputs shapes individual V1 receptive fields. The temporal asymmetry between these pathways was revealed by comparing the neuronal responses to brightened versus darkened stimuli [29].

Recently, using VSD optical imaging, Rekauzke et al showed that the temporal difference in the On-/Off-visual pathways can lead to propagating subthreshold cortical activity, possibly contributing to motion perception [26]. In their experiments, Rekauzke et al used darkening and brightening square stimuli to probe the difference between the On-/Off-visual pathways. The darkening square stimulus is an initially bright square on a uniformly dark background that is changed to grey, and the brightening square stimulus is an initially grey square switched to bright (the leftmost vertical rows in Fig 3A and 3B). VSD imaging captured the cortical

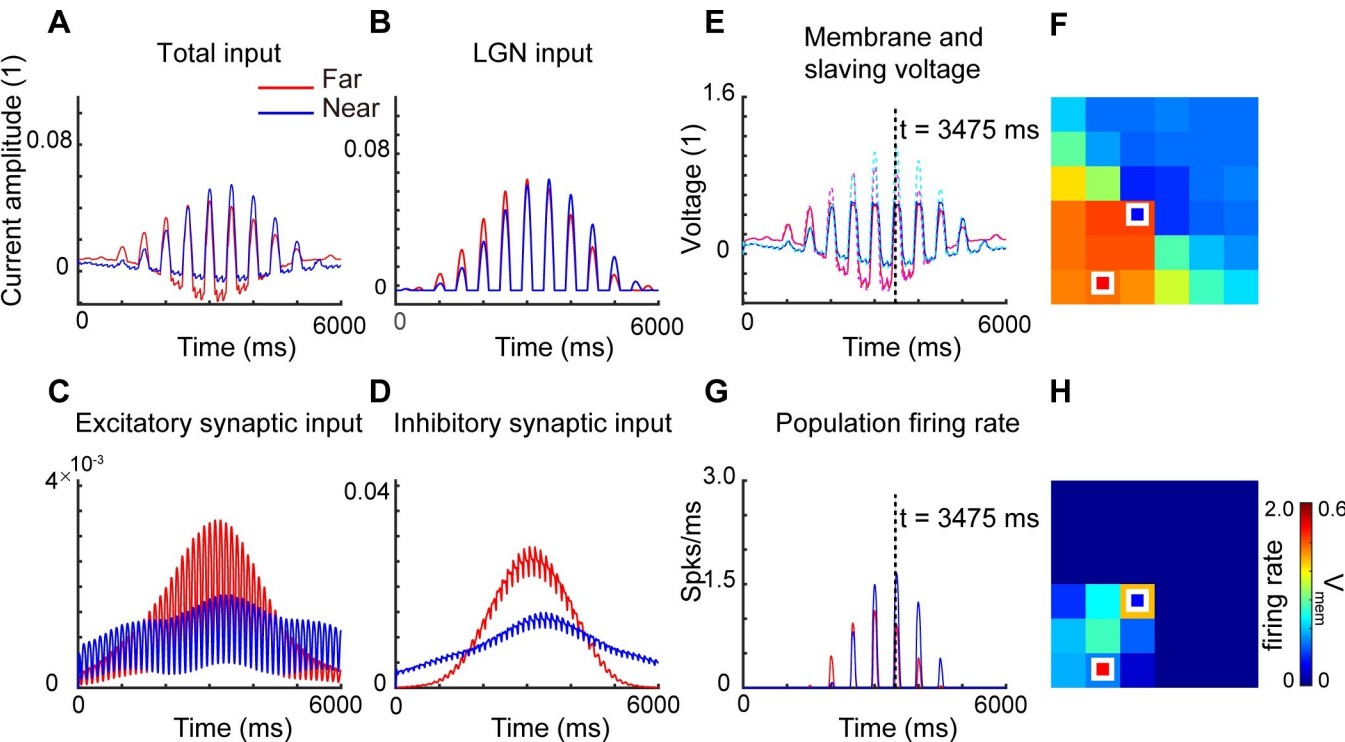

**Fig 2. Responses of our CG moment model to rotating drifting-gratings.** The stimulus is a rotating sinusoidal drifting-grating at the optimal spatial frequency, with a drift rate of 4 Hz per sec and a rotation rate of 20˚per sec. we simulate one single orientation hypercolumn with idealized 'pinwheel' structure for 6000 ms. Overall and constituent current inputs for neurons located far from a pinwheel center and near the pinwheel are summarized. Plotted are A, Total synaptic current inputs, B, external (i.e., LGN) current inputs, C, excitatory cortical current inputs, and D, inhibitory cortical current inputs for the near (blue) and far (red) patch of neurons. E summarizes the temporal dynamics of population-averaged membrane potentials for near (blue) and far (red) neuron patches, and the temporal dynamics of population slaving voltage (total current input divided by leakage conductance) for the same near (cyan dotted) and far (pink dotted) neuron patches. The panel G shown is population firing-rate for near (blue) and far (red) neuron patches. The panels F, H are spatial patterns of population-averaged membrane potential and population firing rate at the same time point (t = 3475 ms).

activities aroused in the directly stimulated location before nearly isotropically spreading via horizontal connections (the top lines of the right panels in Fig 3A and 3B). They also found that responses to the off stimulus (darkened square) arrived ~10 ms before the on stimulus (brightened square), thus confirming the existence of temporal differences between On-/Off-processing.

Using simultaneously darkening and brightening squares at adjacent locations (so-called counterchanging stimulus; see the leftmost vertical row in Fig 3C), Rekauzke et al found propagating cortical activity flowing from the darkened area towards the brightened location. This propagating activity is similar to the wave-like response to a moving square stimulus, suggesting the temporal asymmetries in the On-/Off-visual pathways can lead to motion signals in higher-order visual perception, a hypothesis corroborated by psychophysics experiments.

To generate a physiologically plausible model of the cortex, we use these experimentally recorded VSD data to calibrate our large-scale I&F neuronal network. First, using the results from single darkening and brightening square stimulus, we adjust the LGN input strengths, the On-/Off-LGN temporal kernels, and the strengths of local connectivities. Using the wave propagation properties revealed in the counterchanging stimulus, we calibrate the strengths of long-range connections.

We applied this large-scale I&F model to study the dynamical responses of a realistic cortex, which comprises $\mathcal{O}(10^1)$ hypercolumns and $\mathcal{O}(10^5)$ neurons. Individual neurons and massive,

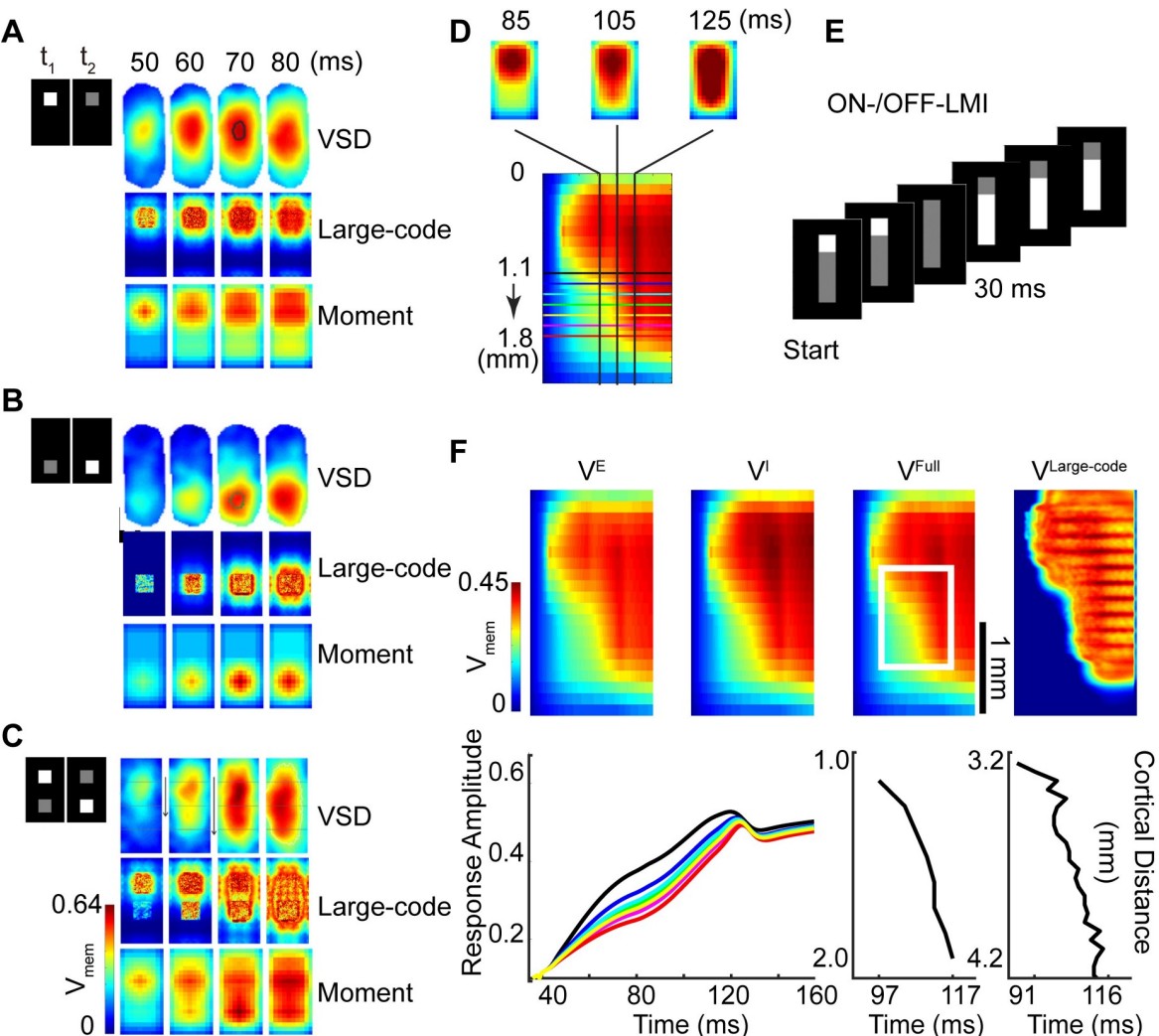

**Fig 3. Comparison of activity patterns between data and models.** Spatiotemporal activity patterns of experimental VSD data (Rekauzke et al JNS 2016), our large-scale integrate-and-fire neuron model and our CG moment model responding to different stimulus paradigms, these stimuli are all involved in On- and Off- temporal processing. For plots **A-C**, left rows, the first and second rectangles represent visual stimuli at two time points, $t_1$ and $t_2$, where $t_2 > t_1$. In plot **A**, the upper square turns dark, in plot **B**, the bottom square turns bright and in plot **C**, the upper square turns dark whereas the bottom square turns bright simultaneously. Right part of each plot, in the first lines, frames show spatiotemporal (10 ms time interval, from 50–80 ms) plots of VSD data, in the second lines, frames show simulated results of our large-scale I&F model, and in the third lines, frames show results of our CG moment model under corresponding stimulus paradigms. **D** describes another spatiotemporal representation for cortical activity pattern, which is derived from frames of two-dimensional spatial activity patterns. These frames are averaged along the horizontal axis (shorter axis) to reduce the 2D spatial patterns to 1D. These 1D patterns are arranged from left to right in time order (i.e., 85ms, 105ms, 125ms 2D spatial activity patterns in top and corresponding 1D vertical lines in the bottom) and aligned in spatial position to obtain the spatiotemporal activity pattern (bottom). Horizontal lines in bottom represent aligned positions and different colors represent different vertical positions. **E**, represents a novel version of the LMI stimulus paradigm, which combines features of the On-/Off-counterchanged square stimulus paradigm of Rekauzke et al. with the standard Hikosaka LMI paradigm, and we call it ON-/OFF-LMI. The stimulus starts with a cue of a small, darkened stationary white square, displays in the first two frames, followed a few milliseconds (about 10–30 ms until the third frame) later by a brightened stationary bar at an adjacent location, which occurs at the fourth frame. **F**, Results of our large-scale and CG moment models under novel ON-/OFF-LMI stimulus paradigm. Top line depicts aggregated and constituent membrane potentials, from left to right, the panel shown are spatiotemporal profiles of population-averaged membrane potentials of an excitatory subpopulation, inhibitory subpopulation, aggregated membrane potentials obtained by CG moment model simulation and aggregated membrane potentials obtained by large-scale I&F model. Left plot in the bottom is time courses of population-averaged responses of CG moment model to ON-/OFF-LMI stimulus paradigm, line colors match the horizontal lines in **D**, represent 7 consecutive adjacent positions spaced at about 0.11 mm intervals within 1.12–1.80 mm, middle panel shows the wave front position (distance from the top of model cortex, 8 adjacent positions from about 1.12 to 1.91 mm) as a function of time (the moment when activity reaches 80% of its maximum amplitude), using results of CG moment model, the velocity is 0.040 = (1.91–1.12)/ (117–97) (mm/ms) [24]. The right panel shows the wave front position of time, but using results of large-scale I&F model, the velocity is 0.038 = (4.2–3.2)/ (116–91) (mm/ms). A color bar that indicates the activity scales is shown in the upper left in (F) and spatial scale is in the right side of the third frame in the identical line.

complex inter-connections are the essential network elements in this model. Although the network dynamics can be accurately traced by a set of ordinary differential equations (ODEs), direct simulations of such a large-scale I&F neuron model is computationally expensive (considering the large number of neurons and the many stochastic realizations of network dynamics that may be required). Furthermore, because of the dimensionality of the network, it is difficult for mathematical analysis.

In matching the dynamical responses of our large-scale I&F model cortex, we found that the illusory phenomenon can be appropriately captured and modeled as the complex collective activities of the cortical circuit, and is crucially dependent on the interaction of neuronal populations. So here, we perform a reduction of the large-scale I&F model, organize neurons with similar properties into spatially coarse-grained subpopulations.

For our coarse-grained, augmented ODE model, we take the parameters directly from the large-scale I&F model. Although a direct comparison between the large-scale I&F model and the CG model shows slightly different spatial activity patterns, the essential wave propagation, from darks to brights, can be reproduced (right panels in Fig 3C). (See Methods for a detailed description for parameter calibration.)

To illustrate our dimensional reduction to capture spatiotemporal cortical activity in general, we stimulate our moment model with a novel version of the LMI stimulus paradigm. In Fig 3E, we combine features of the On-/Off-counterchanging square of Rekauzke et al with the original Hikosaka LMI into a new visual stimulus, which we call the ON-/OFF-LMI. The stimulus starts with a cue of a small, stationary bright square that switches OFF followed a few milliseconds (~10–30 ms) later by a grey stationary bar that turns ON to bright at an adjacent location.

To compare the stimulus-generated neuronal activities between the large-scale I&F network model with our CG results, in Fig 3F, we display activity patterns in a spatially one-dimensional representation (see Fig 3D for details). Excitatory, inhibitory, aggregated population-averaged membrane potentials of patches in the CG model and membrane potentials of point neurons in the large-scale I&F model are displayed from left to right. The membrane potentials initially arise in patches receiving the darkened square stimulus—the earliest responses (cyan/green) appear on the upper left corner of each panel, before spreading isotropically, while the amplitudes gradually increase (yellow/red). In the second stage, after the bar stimulus turns ON to bright, a gradual wave-like propagation of population-averaged membrane potentials emerges. This wave propagation emerges in the region between the middle and bottom portions (white rectangle in Fig 3F) of the activity which receives brightening bar stimulus. The moment in time when the population activities of CG patches reach a particular level (here we chose 80% of the maximum activity) shows a continuous shift in time, i.e., the farther the patch is from the initial stimulus, the later the activity reaches this level. The counterclockwise tilting contours (activities with the same amplitude/color) in the corresponding region intuitively reflect this phenomenon (upper line in Fig 3F). Lower left sub-plot in Fig 3F summarizes the temporal traces of population-averaged membrane potentials at seven evenly spaced locations within the area receiving bar stimulus; we observe a rightward time shift (time delay) along the direction away from the initial stimulus occurring before subthreshold neuronal responses corresponding to the brightening bar stimulus (about 50 ms). Lower right two sub-plots in Fig 3F show the measurement of the propagating wave position as a function of time, from the CG model (left) and large-scale I&F model (right). Both of which have a wave speed of about 0.05 m/s, consistent with experimental results [24].

ON-/OFF-LMI reveals two nontrivial temporal properties: first, the time delay of about 20 ms, inherited from the intrinsic time-latency between On-/Off-visual pathways, and the second is the time difference of about 30 ms between the appearance of the cue-square and the

long bar. The combination of these two timing differences 'primes' the cortical network and initiates a traveling cortical wave when a second stimulus immediately appears in a nearby location [14]. Our CG model indicates that the V1 network can integrate and make full use of these two types of temporal differences, to induce a 'priming' effect, so that an appropriately placed second stimulus triggers cortical voltage propagation. The spatiotemporal activity patterns from the CG model are presented in Fig 3F; as we see, the activities closely resemble the I&F model (rightmost panels in Fig 3F).

We note that the traveling wave is largely independent of the cue contrast. In Fig 4 we plot the trajectory of the propagating wave (in the region marked by white rectangle in Fig 3F) as a function of time. Reducing the darken square's contrast from 100% to 50%, we observe a lagged initiation of the wave. In addition, under these circumstances with different cue contrasts, every point on the trajectory of the traveling wave, shifts by roughly the same period of time (shifts along the x-axis). Fig 4 indicates that after an initial transient, the induced traveling wave reaches a steady state velocity of about 0.05 mm/ms, independent of cue contrast. This speed is roughly consistent with the wave speed induced by the Hikosaka Line-Motion Illusion paradigm.

In Fig 5, we show the CG results from 1) the Hikosaka LMI stimulus (Fig 5A), 2) a moving square (Fig 5D and S1L Fig), 3) two types of drawn-out squares, one directly drawn out to bar length (Fig 5G and S1I Fig), and another starting out as the Hikosaka LMI, with a priming square that vanishes, followed by a drawn-out square (Fig 5J). VSD signals evoked by these stimuli are presented in Fig 5B, 5E and 5H (lacking VSD data for the second type drawn out stimulus); corresponding CG model results are displayed in Fig 5C, 5F and 5I.

We note that in the cases with VSD imaging data, our CG model results reproduce the main features of cortical wave propagation. For instance, in the LMI of Fig 5C, the initial square stimulus activates cortical neurons after about 40 ms, with a persistence of neuronal activity even after the disappearance of the square stimulus (t = 48 ms or 5Δ). Now, at about 60 ms, a bar is flashed. The neuronal patches (VSD or our CG model) in the region near the previous firing build up activities after ~20ms, before spreading to the right. We compare this to a physically moving square stimulus (Fig 5D–5F) to demonstrate that the wave-like cortical activity pattern under LMI is very similar to the activity induced by a square moving at the appropriate speed. The blue line in Fig 5D denotes the trajectory of the head of the moving square stimulus, and the slope of the blue line indicates its speed (~32 deg/sec).

## Discussion

Information processing in the brain is often reflected by organized, coherent activity patterns that are distributed across almost the entire cortex [11, 24, 32, 33]. This population-level neuronal activity is often thought of as an emergent property of strongly coupled recurrent networks [6, 34–36]. Furthermore, it is believed that the information is embedded in the spatiotemporal patterns arising from the collaborations and competitions between the external stimulus, the intrinsic neuronal dynamics and the network architecture. Modern experimental techniques (such as VSD imaging) are capable of capturing this phenomenon with high spatial and temporal resolution [1]. However, the attempt to understand the network mechanisms underlying the generation and maintenance of neuronal population activity has been mainly addressed through large-scale numerical simulations, and a mathematical framework to extract simplified representations remains a major theoretical challenge [15, 28, 35, 37].

The mammalian early visual pathway is a complex system whose functions emerge from interactions that take place simultaneously on a vast range of spatial and temporal scales. Optical imaging experiments of V1 reveal visual feature preferences organized into mm-scale

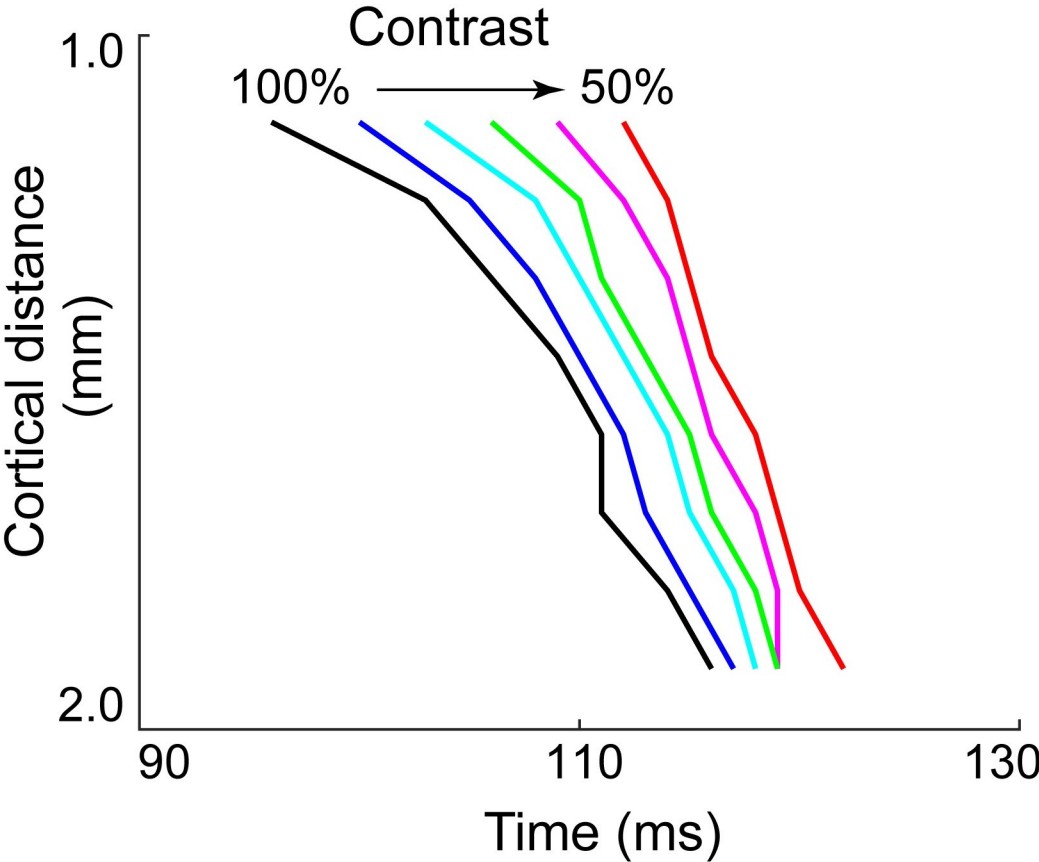

**Fig 4. Effect of cue-contrast dependence on temporal properties of population activity patterns under ON-/OFF-LMI stimulus paradigm.** Y-axis represents the distance from this position to the edge of the model cortex (edge near the position of the stimulus initiation), and x-axis records the moment when the activity of this position reaches 80% of its maximum amplitude. We summarize the results of traveling wave positions as a function of time under identical ON-/OFF-LMI stimulus paradigm, except for the contrast of the initial cue-square, which reduces from 100% to 50% of the contrast fed into the model in Fig 3F. Arrow represents the direction of contrast decreasing in 10% steps (100%/ black, 90%/ blue, 80%/ cyan, 70%/green, 60%/ pink, 50%/ red).

hypercolumns that tessellate the V1 network. While local connectivities on the hypercolumn scale appear to be isotropic, longer range, reciprocal connections are excitatory-only and orientation-specific [38–41]. These long-range horizontal connections are likely to be responsible for the synchronization of gamma-band oscillations that signal visual processing beyond single-cell receptive fields. Voltage-sensitive dye imaging has also revealed the interaction between various spatial and temporal scales to produce visual illusions, such as the line-motion illusion.

Standard ensemble averages of neuronal network dynamics lead naturally to Masters equations, Fokker-Planck systems, and other types of kinetic theories [12, 16, 17, 42–47]. The classic formulation of the neuronal network dynamics on the population level, i.e., population/ ensemble density models or Wilson-Cowan models, evolves according to the conservation of probability density flux and represents the population activity of a cortical region by one or multiple variables.

While these reductions are effective descriptions of coherent population dynamics, they yield systems of partial differential equations that are not always easily amenable to analysis. Mean-field approximations replace the ensemble density with the expected value of the

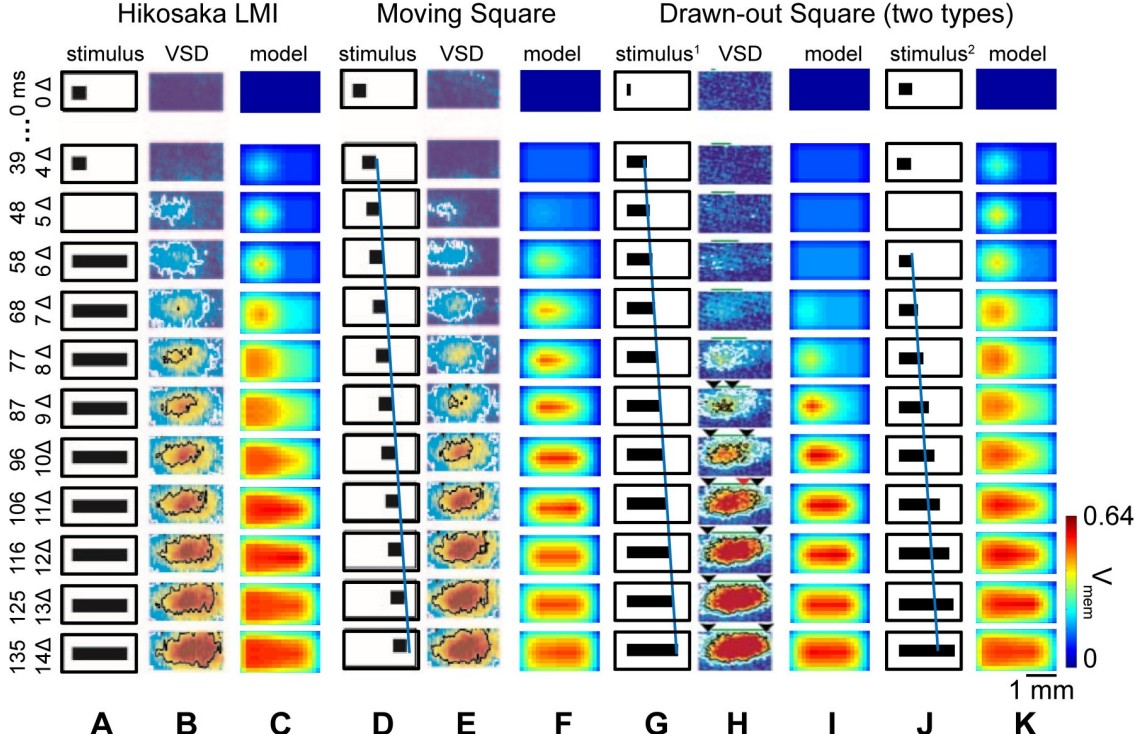

**Fig 5. Comparison of response patterns of CG moment model and experiment under various illusory motion stimulus paradigms.** (A-C) Standard Hikosaka LMI stimulus, (A) visual input, (B) experimental VSD images of cat primary visual cortex, (C) membrane potential pattern of CG moment model are shown in an alignment. (D-F) Moving square stimulus, (D) visual input, blue line denotes the trajectory of the head of this moving square stimulus, the slope reflects its moving speed (1.5˚×3÷140ms≈32˚/sec), (E) experimental VSD images of cat primary visual cortex, (F) membrane potential pattern of CG moment model are shown in alignment. (G-I) The first type drawn-out stimulus, (G) the first type drawn-out visual input, blue line with the same slope as that in D represents the stimulus is drawn out to full bar length at a speed of 32˚/sec, (H) experimental VSD images of cat primary visual cortex, (I) membrane potential pattern of CG moment model are shown in alignment. (J, K) The second type of drawn-out stimulus, (J) the second type drawn-out visual input, this stimulus pattern is identical to Hikosaka LMI in the first 50 ms, then the square stimulus is drawn out to the bar length and the final pattern is identical to the bar stimulus in Hikosaka LMI, (K) membrane potential pattern of CG moment model are shown in alignment. The time dimension is on the left ordinate, the color bar indicates levels of membrane potentials.

network state variables. Further improvements [3, 9] take the fluctuations of neuronal activity into consideration, by introducing *ad hoc* Gaussian noise terms. Recently, a Master equation formalism proposed by Boustani and Destexhe, allows for a 'mesoscopic' level description of population dynamics. Their model can be used beyond stability analysis, but needed to make use of a phenomenologically fitted f-I curve. Other theoretical studies have focused on the dimensional reduction of complex neuronal network exhibiting heterogeneous dynamics. In particular, various studies [16, 48–55] have developed models with conductance moments. This leads to an infinite hierarchy of moments, where the dynamics of lower-order conditional moments depend on higher-order moments, necessitating moment closure assumptions.

Here we use a different method for reduction [19] which was previously shown to be effective for homogeneous I&F networks. In this framework, we obtained a system of ODEs of voltage moments, where the lower-order moments do not depend on higher-order moments. However, at each moment in time, the population firing rate is needed to evolve the system of ODEs. And, in order to compute the population firing rates, we need to compute the full probability distribution function, which cannot be constructed uniquely from a finite set of voltage moments. Therefore, we need a different closure scheme and we choose to use a maximum

entropy assumption. (Technically, we did not maximize the Shannon entropy of the distribution, given the constraints of the time-dependent voltage moments. Instead we maximized the relative entropy between the distribution and the stationary distribution, given the moments. Or, as one of the reviewers pointed out, we minimized the Kullback-Leibler divergence. We are working on understanding the differences in this context and hope to report back soon.)

In this paper, we extend and apply this formalism towards modeling and analyzing large-scale coherent cortical activity in V1. First, we focus on a single V1 orientation hypercolumn before using it to model an extended network that spans roughly 5×3 orientation hypercolumns in V1, an area that contains $\sim \mathcal{O}(10^5)$ neurons. On this scale, VSD imaging of cat V1 revealed coherent wave propagation that may underlie motion perception [26].

While systems of PDEs derived from various kinetic theories [16, 48, 49, 51, 52, 54, 55] are not time consuming to solve, there is a general hope that we can construct a theory of emergent network dynamics in terms of a few dominant, lower order models. Through a maximum entropy voltage moment closure, our dimensional reduction highlights the reduced-dimensional dynamics of the (subthreshold) membrane potential distribution and makes explicit its effects on the dynamics of other neuronal populations. Mathematically, this reduction can be applied to any one-dimensional density distribution. Therefore, we do not need the diffusion approximation (FP equations) *per se* and can incorporate the effects of finite-size synapses [16, 53–56] and other cellular effects, e.g., adaption and short-term depression [7–9]. However, we note that these moment equations simplify greatly in linear I&F networks (a case where all the coefficients of the ODE system can be computed analytically from system parameters). Furthermore, because of the time-scale separation between the fast AMPA and the slow NMDA synapses, asymptotically we can treat the effects of the NMDA synapses as slow currents, keeping our probability distribution function one-dimensional (see S2 Appendix).

In surveying the rich repertoire of cortical dynamics, a natural question arises: what concise, unified characterization can capture the co-existence of diverse, heterogeneous dynamical states in highly recurrent neuronal networks. It is believed that large-scale neuronal information processing emerges from the interaction between the external input, individual neuronal dynamics and cortical architecture. Many studies have been carried out to analyze the dynamical effects of different types of mechanisms [43, 57, 58]. Here we showed how effective a system of voltage moments can be used naturally to model various VSD imaging experiments. Furthermore, our coarse-grained representations can be used to assess the importance of various mechanisms and facilitate our understanding of the rich dynamic states within the mammalian brain. Future work will focus on the incorporation of higher-order network motifs, which are responsible for higher-order correlations beyond the mean-field approximation, and are likely to have important consequences for information processing.

## Materials and methods

### A large-scale V1 model

**An integrate-and-fire neuronal network.**   We model individual V1 neurons (excitatory and inhibitory) as current-based, linear, integrate-and-fire (I&F) point neurons, whose membrane potentials evolve by

$$\frac{d}{dt} V_j^Q = -\frac{1}{\tau_V}\left(V_j^Q - V_R\right) + I_j^{QY} + I_j^{QE} - I_j^{QI}, \tag{1}$$

where the superscript $Q \in \{E, I\}$ represents the type (excitatory or inhibitory) of each neuron, the subscript $j$ indexes the spatial location of the neuron within the V1 network, and $\tau_V = 20$ *ms* is the leakage timescale of the membrane potential. We normalize the membrane potentials

$V_j^Q$ by setting the spiking voltage threshold $V_T = 1$ and the rest (leak) potential $V_R = 0$. In I&F dynamics, the voltage $V_j^Q$ evolves continuously until it reaches a threshold $V_T$ after which it is immediately reset to rest for an absolute refractory period ($t = 2$–$3$ ms). Individual neuron's $V_j^Q$ is driven by its synaptic currents, arising from feedforward input through the LGN ($I_j^{QY}$), and also from recurrent network activities of excitatory ($I_j^{QE}$) and inhibitory populations ($I_j^{QI}$). The I&F neuron model has become a widely-used model for the description of spiking neurons, because of its relative ease for mathematical analysis, and yet its dynamics is sufficiently rich to capture diverse neural processing. The I&F model describes the membrane potential of a neuron in terms of the synaptic current inputs it received, either from cortico-cortical recurrent interactions or from external injections. The various synaptic inputs are as follows:

**The feedforward LGN input.** Our modeling starts with the LGN module. The LGN cells come in two polarities, "On-" and "Off-" cells, each V1 cell receives synaptic inputs from a collection of both On- and Off-LGN cells, with these two types of LGNs segregated spatially into 2D Gabor-like patterns [13, 59, 60]. In the most detailed large-scale I&F model, we randomly sample about $\mathcal{O}(10^1)$ LGN cells within the 2D Gabor envelope and connect them to the same V1 neuron. $N_{lgn}$ LGN cells connected to the $j^{th}$ V1 neurons create a sequence of spikes at times $\{T_{i,k}^Y\}$, which is a Poisson process. Each spike causes a synaptic current of synaptic strength $S^{QY}$, so then the total external LGN input $I_j^{QY}$ can be modeled as,

$$I_j^{QY} = S^{QY} \sum_{i=1}^{N_{lgn}} \sum_k \alpha_{ext}(t - T_{i,k}^Y) = S^{QY} \sum_{i=1}^{N_{lgn}} \sum_k \delta(t - T_{i,k}^Y). \tag{2}$$

The temporal kernel $\alpha_{ext}$ is an $\alpha$-function and models the time course of the synaptic current induced by each LGN spike (each $T_{i,k}^Y$) [60, 61]. In the following coarse-graining reduction, the time scale of $\alpha_{ext}$ is assumed to be infinitely fast and model each spike as a delta function, $\delta(t - T_{i,k}^Y)$. (Here we are explicitly modeling phenomena slower than AMPA and GABA but faster than NMDA time-scales.)

Our first coarse-graining approximation is to use continuous rate $\eta_j^{ext}(t)$ rather than discrete Poisson spikes $\delta(t - T_{i,k}^Y)$ to describe the total external input into the $j^{th}$ V1 neuron arrives from $N_{lgn}$ LGN cells [13, 62], so we let

$$I_j^{QY} = S^{QY} \cdot \eta_j^{ext}(t). \tag{3}$$

According to different spatiotemporal patterns of visual stimuli, the continuous, time-dependent rate of the Poisson process $\eta_j^{ext}(t)$ varies.

For the rotating drifting grating stimulus used in the first experiment, the visual stimulus is an intensity pattern $I(\mathbf{X}, t)$ given by

$$I(\mathbf{X}, t) = I_0[1 + \epsilon \sin(\mathbf{k} \cdot \mathbf{X} - \omega t + \varphi)], \tag{4}$$

where $\varphi$ describes the stimulus spatial phase and $\mathbf{k} = |k|(\cos\theta_t, \sin\theta_t)$ reflects the spatial frequency and instantaneous orientation at time $t$. The response of each cell in the LGN module can be modeled as a rectified, linear spatiotemporal convolution of the visual stimuli, where spatial and temporal kernels are constrained by experiments[42, 59]. Then, following Shelley & McLaughlin [61], the total, continuous current input into the $j^{th}$ cortical neuron is approximated by

$$\eta_j^{ext} = C\epsilon\left[1 + \frac{1}{2}(1 + \cos 2(\theta_j - \theta_t))\sin(\omega t - (\varphi_j - \varphi))\right], \tag{5}$$

where $\theta_j$ reflects preferred orientation and $\varphi_j$ the preferred phase of the $j^{th}$ cortical neuron. Note that, in the large-scale I&F model, the index $j$ labels single neuron, but in the coarse-grained network model (see below), we assume the V1 neurons within one coarse-grained patch are homogeneous, so the index $j$ can also index each individual neurons in the $j^{th}$ homogeneous patch.

In our second experiment, transient On- and Off-visual stimuli were used to probe the dynamics of the On- and Off-visual pathways. The On- and Off-visual pathways already exhibit differences at the LGN [63–66], such as spatially segregated On- and Off- afferent couplings from the RGCs, different response times of the On- and Off-visual pathways and so on. Here we use an $\alpha$-function to describe the total, temporal responses of a collection of LGN cells to transient On- and Off-stimuli. So, the continuous, external current input under this condition is

$$\eta_j^{ext,On(Off)}(t) = \frac{t}{\tau_1^2}\exp\left(-\frac{t}{\tau_1}\right) - \frac{t}{\tau_2^2}\exp\left(-\frac{t}{\tau_2}\right). \tag{6}$$

We use two sets of parameters ($\{\tau_1,\tau_2\}$ in Eq (6)) for $\eta_j^{ext,On}$ and $\eta_j^{ext,Off}$ to model the On- and Off-LGN feedforward time courses, respectively (Fig 1B).

**Cortical Architecture.** To study the dynamics of a patch of a layer of V1, we construct a 2-dimensional network with spatially structured synaptic connections, through which V1 excitatory and inhibitory neurons recurrently interact. These recurrent cortical connections are represented by the third and fourth terms in Eq (1).

In our first experiment, a single orientation hypercolumn with multiple orientation domains was modeled and only local cortical interactions (<500 μm) were included in the simulations. Therefore, we regard the network as an idealized two-dimensional neuronal network, with all-to-all, isotropic cortical connections. The strengths of these synaptic couplings fall off as the spatial separation between two neurons [67–69].

In the more complicated, second experiment, we model a larger patch of V1, with a spatial range of 2.5×1.5 mm² and containing 5×3 orientation hypercolumns. On this scale, it is crucial to include long-range synaptic connections in the model. The strengths of long-range (>1000 μm) horizontal synaptic connections specifically depend on orientation preferences of the pre- and post-synaptic neurons [38, 40, 41]:

$$\begin{aligned}
I_j^{QE} = &\sum_{j'\in E}\sum_k S_{fast}^{QE}K_{AMPA}^{QE}(|c_j - c_{j'}|)\alpha_{AMPA}(t - T_{j',k}^E) \\
&+ \sum_{j'\in E}\sum_k S_{slow}^{QE}K_{NMDA}^{QE}(|c_j - c_{j'}|)\alpha_{NMDA}(t - T_{j',k}^E),
\end{aligned} \tag{7}$$

$$I_j^{QI} = \sum_{j'\in I}\sum_k S_{fast}^{QI}K_{GABA}^{QI}(|c_j - c_{j'}|)\alpha_{GABA}(t - T_{j',k}^I), \tag{8}$$

where the excitatory and inhibitory synaptic currents have the form (Eqs 7 and 8), where $Q\in\{E,I\}$. Here $T_{j',k}^{E/I}$ denotes the time of the $k_{th}$ spike of the $j'_{th}$ excitatory/ inhibitory neuron. We include slow NMDA synaptic currents in addition to the fast excitatory synaptic currents mediated by AMPA and the fast inhibitory synaptic currents mediated by GABA [1, 38, 39, 41, 46]. The normalized spatial profile of the cortical coupling strengths ($K^{QQ'}(d)$), i.e., both short-range local connections and long-range horizontal connections, are modeled as normalized 2D Gaussian functions (Eq (9)) of the cortical distance between two neurons (or two coarse-graining populations) $d = |c_j - c_{j'}|$. $L_P^{QQ'}$ denotes the spatial length-scale of the corresponding

type of connections. The spatial kernels are normalized by $A_P^{QQ'}$, as described in previous studies [67, 70]

$$K_P^{QQ'}(d) = A_P^{QQ'} \exp\left(-\frac{d^2}{(L_P^{QQ'})^2}\right). \tag{9}$$

Time courses due to individual spikes at the time $\{T_{j',k}\}$ of the cortical recurrent input can be expressed in the form of alpha functions,

$$\alpha_P(t) = A_{max} B\left(\exp\left(\frac{-(t - T_{j',k})}{\tau_1}\right) - \exp\left(\frac{-(t - T_{j',k})}{\tau_2}\right)\right), \tag{10}$$

$$B = \left(\left(\frac{\tau_2}{\tau_1}\right)^{\tau_r/\tau_1} - \left(\frac{\tau_2}{\tau_1}\right)^{\tau_r/\tau_2}\right)^{-1}, \tag{11}$$

where $\alpha_P(t)$, $P \in \{$AMPA, NMDA, GABA$\}$, $\tau_1 > \tau_2$ and $B$ is a normalization factor that assures the peak value $A_{max}$, with a different rise- ($\tau_r = \tau_1\tau_2/(\tau_1-\tau_2)$) and decay- ($\tau_d = \tau_1$) time constants (Eqs 10 and 11). In our theoretical model below, we model the synaptic time courses of fast excitatory (AMPA) and inhibitory (GABA) synapses with an instantaneous rise-time and an infinitely fast decay-time. Therefore, once a cortical neuron fires, the fast synaptic cortical currents create an instantaneous jump in the membrane potential of the post-synaptic neuron. We model the effect of the slow NMDA-type current with nearly instantaneous rise-time $\tau_{r,NMDA}^E \approx 2$ ms and a decay-time $\tau_{d,NMDA}^E \approx 80 - 130$ ms. Thus, the recurrent, cortical synaptic inputs are given by

$$I_j^{QE} = I_{j,fast}^{QE} + I_{j,slow}^{QE}, \tag{12}$$

$$I_{j,fast}^{QE} = \sum_{j' \in E} \sum_k S_{fast}^{QE} K_{AMPA}^{QE}(|c_j - c_{j'}|)\delta(t - T_{j',k}^E), \tag{13}$$

$$I_{j,slow}^{QE} = \sum_{j' \in E} \sum_k S_{slow}^{QE} K_{NMDA}^{QE}(|c_j - c_{j'}|)\alpha_{NMDA}(t - T_{j',k}^E), \tag{14}$$

$$I_{j,fast}^{QI} = \sum_{j' \in I} \sum_k S_{fast}^{QI} K_{GABA}^{QI}(|c_j - c_{j'}|)\delta(t - T_{j',k}^I). \tag{15}$$

Fast/slow excitatory recurrent synaptic strengths associated with excitatory- and inhibitory-type postsynaptic neurons are $S_{fast/slow}^{EE}$ and $S_{fast/slow}^{IE}$, respectively. Similarly, the fast-inhibitory synaptic strengths associated with excitatory- and inhibitory-type postsynaptic neurons are represented by $S_{fast}^{EI}$ and $S_{fast}^{II}$. Because all spatial kernels are normalized, these parameters label the strengths and relationships of synaptic couplings.

Thus, our model is an idealization of the experimentally observed connectivities of a V1 layer, the strengths and spatiotemporal properties of the synaptic connections depend on the neuronal types, on orientation preferences, and whether they lie in the same orientation hypercolumn.

**A coarse-grained neuronal network.** Intensive studies have suggested that cortical functional maps, such as orientation pinwheel structure, phase preference, spatial frequency preference, are arranged in regular, organized spatial patterns across visual cortex. Therefore, we partitioned the 2D cortical network into small patches, each of which is large enough to

contain a large number of neurons, but still small enough to ensure that functional (i.e., physiological) properties, like orientation preferences, are roughly constant within one single patch.

Specifically, in the first experiment, only one hypercolumn was simulated to model the cortical responses to rotating drifting-grating. In the primary visual cortex, the cortical responses of cells show sensitivity to orientation through elevated firing rates, and the spatial phases, which depend on the position of the grating stimulus. Orientation preference is arranged in pinwheel while spatial phase preference is distributed randomly. To ensure neuronal populations contain disordered, well-sampled preferred spatial phases, we designed 4 similar cortical patches covering every single hypercolumn, but with different preferred spatial phases (0˚, 90˚, 180˚, 270˚) (see Fig 1). Each hypercolumn has regular pinwheel-structured orientation map and further coarse-grained (CG) into 6×6 CG patches. Individual CG patch with a particular preferred spatial phase consists of 58 neurons, these neurons located in clusters and held similar orientation preference. Furthermore, single hypercolumn simulated in this model suggested that only local cortical interactions ($<$500 μm) were included in the model, and these interactions were all-to-all connected and assumed to be isotropic.

In the second experiment, a larger cortical area comprising many hypercolumns, was modeled. Thus, the synaptic connections of this simulating model consisted of isotropic short-range connections and long-range horizontal synaptic connections, which were known to depend on orientation preference. In order to capture the population dynamics of this large cortical area and emphasize the significant role of long-range orientation-dependent connections, we structured our model, to divide each hypercolumn into 4×4 CG patches. According to their positions in the pinwheel-structured orientation map, these CG patches belonged to 4 different orientation clusters. In addition to local isotropic short-range connections similar to those in the first experiment, long-range horizontal connections across different hypercolumns were also considered. In summary, this model had 5×3 hypercolumns, each of which further divided into 4×4 CG patches. The specific space-time settings of population dynamic framework studied in this work, are shown in Fig 1.

**Coarse-grained network model.** Once the full I&F model network configuration is set up, we can start to coarse-grain. As it is commonly used in population density methods, we consider two biophysically relevant mesoscopic quantities–the firing-rate and the distribution of neuronal membrane potentials.

The voltage distribution $\rho_j^Q(v, t)$ of finding a neuron whose membrane potential is in voltage bin ($v,v+dv$) at time point $t$ within a given ensemble (labeled by $j$ and $Q$), is governed by a Master equation:

$$\frac{d\rho_j^Q(v,t)}{dt} = g_L \partial_v \left[ \left( v - V_R - \frac{I_{j,slow}^{QE}(t)}{g_L} \right) \rho_j^Q(v,t) \right] + \eta_j^Q \left[ \rho_j^Q(v - f_j^Q, t) - \rho_j^Q(v,t) \right]$$
$$+ N_E \sum_i m_i^E(t) [\rho_j^Q(v - S_{fast}^{QE} K_{fast}^{QE}(|c_j - c_i|), t) - \rho_j^Q(v,t)] \qquad (16)$$
$$+ N_I \sum_i m_i^I(t) [\rho_j^Q(v + S_{fast}^{QI} K_{fast}^{QI}(|c_j - c_i|), t) - \rho_j^Q(v,t)],$$

where $m^{E/I}(t)$ is the excitatory and inhibitory population-averaged mean firing rate of each neuron as a function of time. Note that this equation has already related the two relevant mesoscopic quantities, firing rate and distribution of membrane potential.

We first focus on the evolution of voltage probability distribution induced by various synaptic inputs. Therefore, we define the probability flux $J_j^Q[\rho_j^Q(v,t)]$, which, conventionally, represents the probability crossing a voltage point $v$ at time point $t$. In the Master equation Eq

(16), there are one type of streaming flux due to the relaxation dynamics and another type of spike-driven flux induced by voltage jumps once receiving an input spike. Various researchers have utilized the Master equation combining with the probability flux to analytically or numerically solve the evolution of voltage probability density and treat the network dynamics (e.g., [16, 50, 55, 56]).

For further simplification and theoretical analysis, we try to do some rational approximations. We assume that the voltage jumps due to the external Poisson spikes as well as the recurrent network spikes are small, so the Master equation Eq (16) can be approximated by a standard Fokker-Planck type equation

$$\frac{d\rho_j^Q(v,t)}{dt} + g_L\partial_v J_j^Q[\rho_j^Q(v,t)] = 0, \text{ for } v \in [V_R, V_T), \tag{17}$$

where the probability flux $J_j^Q[\rho_j^Q(v,t)]$ is of a simpler form,

$$J_j^Q[\rho_j^Q(v,t)] = -(v-\mu_j^Q)\rho_j^Q(v,t) - \frac{(\sigma_j^Q)^2}{2}\partial_v\rho_j^Q(v,t), \tag{18}$$

with drift and diffusion terms. The drift and diffusion coefficients $\mu_j^Q$ (also called slaving-voltage) and $\sigma_j^Q$ depend on the spatiotemporal network couplings as well as cortical activities $m_i^{E/I}$, and can be written as

$$\mu_j^Q = (V_R g_L + f_j^Q \eta_j^Q + N_E S_{fast}^{QE}\sum_i K_{fast}^{QE}(|c_j - c_i|)m_i^E$$
$$-N_I S_{fast}^{QI}\sum_i K_{fast}^{QI}(|c_j - c_i|)m_i^I + I_{j,slow}^{QE})/g_L, \tag{19}$$

$$(\sigma_j^Q)^2 = ((f_j^Q)^2\eta_j^Q + N_E\sum_i (S_{fast}^{QE}K_{fast}^{QE}(|c_j - c_i|))^2 m_i^E$$
$$+N_I\sum_i (S_{fast}^{QI}K_{fast}^{QI}(|c_j - c_i|))^2 m_i^I)/g_L. \tag{20}$$

The detailed derivations are in S1 Appendix, and some relevant references can be found in [16, 19, 51, 52, 71].

Now we discuss the boundary conditions. Note that $\rho_j^Q(v,t) = 0$ when $(v \to -\infty)$ and the neuron can only fire by receiving an excitatory spike and cannot stream up to cross the voltage threshold. A negative flux at threshold $V_T$ is impossible and it can only reset once it arrives at the voltage threshold. Therefore, we have boundary conditions

$$\rho_j^Q(V_T, t) = \rho_j^Q(-\infty, t) = 0. \tag{21}$$

We then consider a transient reset dynamics of the I&F type neurons in our model. Once the voltage crosses the voltage threshold $V_T$, it immediately resets to rest voltage $V_R$ without any refractory-period. Therefore, to ensure the continuity of voltage distribution, we have

$$J_j^Q[\rho_j^Q(V_T, t)] = J_j^Q[\rho_j^Q(V_R^+, t)] - J_j^Q[\rho_j^Q(V_R^-, t)]. \tag{22}$$

One of the most important statistical characterizations of neuronal networks is the firing rate, which reflects the suprathreshold network dynamics. The firing rate $m_j^Q(t)$ is related to

the probability through the spiking threshold, which is given by

$$m_j^Q(t) = g_L J_j^Q[\rho_j^Q(V_T, t)] = -\frac{g_L (\sigma_j^Q)^2}{2} \partial_v \rho_j^Q(V_T, t) \tag{23}$$

under the boundary condition $\rho_j^Q(V_T, t) = 0$. Thus, using the continuity of the voltage probability distribution Eq (22), we then have

$$J_j^Q[\rho_j^Q(V_R^+, t)] - J_j^Q[\rho_j^Q(V_R^-, t)] = \frac{m_j^Q(t)}{g_L}. \tag{24}$$

In the evolution of this system, we can determine the firing rate $m_j^Q(t)$ via Eq (23) once we know the solution of the full probability distribution function $\rho_j^Q(v, t)$. However, (Eqs 17–20) show that the voltage probability distribution function is specified with $\mu_j^Q$ and $\sigma_j^Q$, which are, in turn, functions of the firing rate $m_i^Q(t)$. Therefore, Eq (17) is a nonlinear partial differential equation.

As we showed previously, further dimensional reduction can be achieved [27].

First, let us define the $k^{th}$-order voltage moments $\chi_j^{Q,k}(t)$ of $\rho_j^Q(v, t)$ by:

$$\chi_j^{Q,k}(t) = \int_{-\infty}^{V_T} v^k \rho_j^Q(v, t) dv. \tag{25}$$

According the definition Eq (25) of the voltage moments, multiplying $v^k$ by both sides of the original Fokker-Planck equation Eq (17), and taking integration by part over interval $(-\infty, V_T)$, we then have that the voltage moments $\chi_j^{Q,k}(t)$ evolve over time as

$$\frac{d}{dt} \chi_j^{Q,k}(t) = -m_j^Q(t) - g_L[\chi_j^{Q,k} - \mu_j^Q] \text{ for } k = 1, \tag{26}$$

$$\frac{d}{dt} \chi_j^{Q,k}(t) = -m_j^Q(t) V_T^k - kg_L \left[ \chi_j^{Q,k} - \mu_j^Q \chi_j^{Q,k-1} - \frac{k-1}{2}(\sigma_j^Q)^2 \chi_j^{Q,k-2} \right] \text{ for } k > 1. \tag{27}$$

The evolution of the ODE system (Eqs 26 and 27) depends on $\mu_j^Q$, $\sigma_j^Q$ and the firing rate $m_j^Q(t)$, but does not explicitly depend on the higher-order moments. Noting that the computation of the firing rate from Eq (23) requires the information of full voltage probability $\rho_j^Q(v, t)$. And the $\rho_j^Q(v, t)$ cannot be determined exactly by the full-moments, hence we say the ODE system (Eqs 26 and 27) is not closed. To close the system of ODEs, we now use a maximum-entropy solution to approximate $\rho_j^Q(v, t)$ (more details are given in S4 Appendix). For completeness, we briefly review the basic idea of the maximum-entropy formulation as follows.

For the given a finite set of moments $\{\chi_j^{Q,k}(t)\}$, $k = 1, 2, \ldots, M$ and the terms $\mu_j^Q$ and $\sigma_j^Q$ on the right-hand-side of ODEs (Eqs 26 and 27), there are many possible voltage distributions $\rho_j^Q(v, t)$ which are compatible with the entire information we have in, i.e., $\{\chi_j^{Q,k}(t)\}$, $\mu_j^Q$ and $\sigma_j^Q$. From amongst all possibilities, we choose the one that lies closest to the equilibrium solution $\rho_{j,Eq}^Q(v)$ that the system would adopt. The derivation of the equilibrium solution $\rho_{j,Eq}^Q(v)$ for given $\mu_j^Q$ and $\sigma_j^Q$ is in S3 Appendix.

To determine how 'close' any particular voltage distribution $\rho_j^Q(v, t)$ is to the stationary solution $\rho_{j,Eq}^Q(v)$, we turn to the dynamics of Eq (17). If we were to assume that $\mu_j^Q$ and $\sigma_j^Q$ were fixed, we have the equilibrium solution of Eq (17), given in Eq.(C.3) in S3 Appendix. More specifically, we can reconstruct the 'most likely' underlying distribution $\rho_j^Q(v, t)$ by maximizing

the entropy:

$$\text{maximize } H(\rho_j^Q) = -\int_{-\infty}^{V_T} \rho_j^Q(v, t) log\left(\frac{\rho_j^Q(v, t)}{\rho_{j,Eq}^Q(v)}\right) dv,$$

$$\text{subject to } \chi_j^{Q,k}(t) = \int_{-\infty}^{V_T} v^k \rho_j^Q(v, t) dv. \tag{28}$$

Set $\lambda_j^Q = [\lambda_{j,0}^Q, \lambda_{j,1}^Q, \ldots, \lambda_{j,M}^Q]^t$ to be a set of Lagrange multipliers for the $j^{th}$ population, corresponding to $M$-order moments. The solution to the optimization problem described by Eq (28) can be written as

$$\rho_j^Q(v, t) = \rho_{j,Eq}^Q(v) \exp(\sum_{k=0}^{M} \lambda_{j,k}^Q v^k - 1), \tag{29}$$

which is the most probable voltage distribution we need. Thus, from Eq (23), we can use $\rho_j^Q(v, t)$ to calculate the firing rate $m_j^Q(t)$ at time $t$ by

$$m_j^Q(t) = -\frac{g_L(\sigma_j^Q)^2}{2} \partial_v \rho_j^Q(V_T, t) = \frac{g_L \sigma_j^Q}{2C} \exp(\sum_{k=0}^{M} \lambda_{j,k}^Q V_T^k - 1), \tag{30}$$

where $C$ is the normalization factor.

Combining with Eq (30), the ODE system (Eqs 26 and 27) has a closure with the maximum entropy principle. This ODE system is also called by an augmented ODE system in the previous works [19, 27, 71]. We found that in the simulations performed here, good performance can be achieved with only 4 augmented variables $\{\chi_j^{E,1}(t), \chi_j^{E,2}(t), \chi_j^{I,1}(t), \chi_j^{I,2}(t)\}$ for each CG patch.

**Parameter calibration.**   First, we choose the strength of external, feedforward input (i.e., LGN input) based on observations under a single square stimulus. Single darkening/brightening stimulus generates moderate population activity (i.e., subthreshold membrane potential and suprathreshold firing-rate), we specifically change the external driving strength $S^{QY}$ so that the voltage component generated by the external input is greater than the voltage threshold, which means that the external input can trigger population firing event in the directly stimulated regions at the initial stage. This initial population firing event is one of the conditions for the succeeding isotropically spreading activity. Second, we choose a larger strength for excitatory local connection than inhibitory one, to ensure the local spreading (recruitment). Furthermore, based on the observed differences in the initial time of the responses to On-/Off-stimulus, we select different time parameters $\{\tau_1, \tau_2\}$ (Eq (5)) to model the different feedforward time kernels of the On-/Off-visual pathways.

Considering experimentally observed VSD data under visual stimulus with more complex spatiotemporal structure, i.e., counterchanging On-/Off-stimulus, Hikosaka LMI stimulus, the propagating wavelike population activity pattern suggests the critical role of long-range NMDA-type synaptic connection [72, 73]. We then choose the strengths of the long-range connections within our model, so that our simulation results could reproduce the traveling wave population activity pattern, qualitatively matching the wave speeds in the experimental data.

## Supporting information

**S1 Appendix. A. Derivation of Fokker-Planck equation.**
(DOCX)

**S2 Appendix. B. Reduction process for the slow current $I_i^s$.**
(DOCX)

**S3 Appendix. C. Derivation of the stationary solution $\rho_{Eq}$.**
(DOCX)

**S4 Appendix. D. Maximum Entropy approximation.**
(DOCX)

**S1 Additional Evidence. (Additional supporting evidence for the Results).**
(DOCX)

**S1 Fig. Response patterns of CG moment model under other stimulus paradigms and using other parameter sets.** (L, M) Moving square stimulus moving at another speed, (L) visual input, which is identical to the previous one shown in Fig 5D except for a slower moving speed, red line denotes this slower moving speed of about 20˚/sec, blue dash-line represents the previous moving speed 32˚/sec, (M) membrane potential pattern of CG moment model. (I, O) The first type drawn-out stimulus, stimulus is drawn out to full bar length at another speed, (I) visual input, which is identical to the previous one shown in Fig 5G except for a slower drawn-out speed, red line denotes this slower speed of about 20˚/sec, (O) membrane potential pattern of CG moment model. (P, Q) Reversed drawn-out stimulus, (P) visual input, this stimulus paradigm is the same as Hikosaka LMI in the initial time period, but after 60 ms, it initiates from the right area and is drawn inward, (Q) membrane potential pattern of CG moment model. (R, S) Hikosaka LMI paradigm stimulates CG moment model with very strong local inhibition, (R) visual input (Hikosaka LMI stimulus paradigm), (S) membrane potential patterns of CG moment model with strong local inhibition, the new local inhibitory connections $S_{new}^{QI} = S_{old}^{QI} \times 1E1 \gg S^{QE}$. (T-V) Single bar stimulus, (T) visual input, (U) experimental VSD images of cat primary visual cortex, (V) membrane potential pattern of CG moment model (original).
(TIF)

**S2 Fig. Another spatiotemporal diagram and more results of simulation responses under different stimulus paradigms.** (A) Response to Hikosaka LMI stimulus, the first line shows spatiotemporal diagrams of population-averaged membrane potential of excitatory subpopulation (left), inhibitory subpopulation (middle) and aggregated result (right), Left plot in the bottom is time courses of population-averaged membrane potentials (same conventions as in the bottom left plot of Fig 3F), right panel shows the wave position as a function of time (same conventions as in the bottom middle and right plots of Fig 3F), the velocity is 0.042 = (1.91–1.12)/ (109–90) (mm/ms). (B) Responses to moving square stimuli at three different speeds. The first line shows spatiotemporal diagrams of population-averaged membrane potential under moving square stimulus at a speed of 64˚/*sec* (left), 32˚/*sec* (middle, corresponding to Fig 5D–5F), and 20˚/*sec* (right, corresponding to S1L Fig), the bottom plot summarizes the temporal functions of wave position at a speed of 64˚/*sec* (black), 32˚/*sec* (blue), and 20˚/*sec* (red), corresponding result under Hikosaka LMI (green dash-line) is plotted for comparison. (C, D) Results under the first type drawn-out stimuli, (C) stimulus is drawn-out at a speed of about 32˚/*sec* (corresponding to Fig 5G–5I), the velocity of traveling wave is 0.049 = (1.91– 1.12)/ (126–110) (mm/ms) (D) stimulus is drawn-out at a speed of about 20˚/*sec*

(corresponding to S1I and S1O Fig), the velocity of traveling wave is 0.029 = (1.91–1.12)/(149–122) (mm/ms). (E) Results under reversed drawn-out stimulus (corresponding to S1P and S1Q Fig). (F) Results under Hikosaka LMI stimulus, using CG moment model with strong inhibition (corresponding to S1R and S1S Fig). Detailed descriptions for each subplot in (C-F) are in the same conventions as in (A). Color bar and spatial scales are in the same conventions as in Fig 3F.
(TIF)

## Acknowledgments

We thank Dajun Xing and Yuhan Chen for discussions.

## Author Contributions

**Conceptualization:** Yuxiu Shao, Jiwei Zhang, Louis Tao.

**Formal analysis:** Yuxiu Shao, Jiwei Zhang, Louis Tao.

**Funding acquisition:** Jiwei Zhang, Louis Tao.

**Investigation:** Yuxiu Shao, Louis Tao.

**Methodology:** Yuxiu Shao, Jiwei Zhang.

**Visualization:** Yuxiu Shao, Jiwei Zhang.

**Writing – original draft:** Yuxiu Shao, Jiwei Zhang, Louis Tao.

**Writing – review & editing:** Yuxiu Shao, Jiwei Zhang, Louis Tao.

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
