## [Decision Letter · Decision Letter 0]

1 Oct 2019

Dear Dr Tao,

Thank you very much for submitting your manuscript 'Dimensional Reduction of Emergent Spatiotemporal Cortical Dynamics via a Maximum Entropy Moment Closure' for review by PLOS Computational Biology. Your manuscript has been fully evaluated by the PLOS Computational Biology editorial team and in this case also by independent peer reviewers. The reviewers appreciated the attention to an interesting problem, but two of them raised substantial concerns about the manuscript which are sufficiently adverse that we cannot accept your paper on the basis of material now at hand. If you feel that you can overcome or refute the criticism, we are willing to consider a revised version in which the issues raised by the reviewers have been adequately addressed.

Sincerely,

Ernest Montbrió, Ph. D.

Guest Editor

PLOS Computational Biology

Lyle Graham

Deputy Editor

PLOS Computational Biology

[LINK]

**Comments to the Authors:**

Reviewer #1: I think this paper is fine. Perhaps they can give more details or cite the relevant reference from how they get from (15) to (16).

The resulting reports include critiques that Unfortunately, two of the reports

Reviewer #2: The authors model published VSD data in a local network of V1 populations, using a reduced population density model obtained by moment closure.

One problem is that although the model replicates the data, the data appear not that challenging to model. The authors provide a plausible interpretation of the experimental data and provide a reasonable interpretation of the neural circuits that can generate them. But overall they seem more concerned with the internal workings of their model than with the data. Could these data have been equally well modeled with Wilson-Cowan dynamics? I’d expect so, and it would have been a relatively straightforward exercise for them to have tried this, after putting in the work to model the architecture. If that is the case, what is the value of their method for this model?

I’m surprised that earlier work on moment closure was not cited, e.g. (Ly & Tranchina, 2007). How does the authors’ work differ from that?

The authors simply refer to their earlier paper for details on their method and while I understand that they don’t want to repeat the full analysis here, there are certain non-trivial elements to this approximation that should have been discussed here. The introduction of fast and slow synaptic currents (NMDA times, line 179-180) would suggest a non trivial exercise because this would lead to a two dimensional density at least (membrane potential and conductance state). However, the Fokker-Planck equation is simply one dimensional (Eq. 16) and appears to be the standard population density equation. Does the closure method work on a 1D density?

Why do the authors assume synaptic efficacies are small? There is no need to rely on that assumption: Nykamp & Tranchina (2000), de Kamps (2003), Iyer et al (2013) give efficient algorithms for finite-size synapses. Experimental evidence that the diffusion approximation is warranted has not been presented by the authors. Is the moment closure method critically dependent on it?

The authors state that two to four moments are sufficient to represent the density. This suggests that the density is never far from equilibrium, in particular if only two moments suffice. The authors could have implemented the network using the Fokker-Planck equation and compare the density spanned by the two (or four) moments with that of the FP equation. This would have shown something about how well their method performs.

The authors are correct that a well motivated reduction of FP equations to ODEs is computationally efficient for large networks and worthwhile pursuing. To make that case, in my opinion, they should have modeled a network that is not described well by rate based equations (Wilson-Cowan), but that is by FP (or other population based methods) as well as by their method.

There are other reductions two ODE-like systems, such as that Mattia & del Giudice. Although they were obtained by different methods, it is not clear to me that the approach describes here represents a major advance in the state-of-the-art.

Reviewer #3: The manuscript "Dimensional Reduction of Emergent Spatiotemporal

Cortical Dynamics via a Maximum Entropy Moment Closure" employs a

coarse-graining technique based on a population-density approach

combined with an approximate moment closure to obtain a set of

coarse-grained equations of motion for networks of spiking neurons.

In particular the authors here investigate the application of their

method to large-scale models of V1 with realistic features in terms of

connectivity.

The manuscript is potentially an intersting contribution to the field.

However, I have some major concerns about the employed approximations

that need to be clarified before I would recommend publication in

PloS CB.

Major:

1.

The method proposed by the authors intends to perform a

coarse-graining of leaky integrate-and-fire models. It rests on a

maximum entropy closure of the moment hierarchy. To this end, the

authors employ the equilibrium solution of the Fokker-Planck equation

to find the expansion of the density in a number of moments that

matches most closely the equilibrium distribution.

This scheme seems to be inconsistent with the with to obtain a

coarse-grained theory for transient dynamics, because the equilibrium

distribution, their eq. (22), is of course not a solution of the

time-dependent problem.

The authors should explain how this approximation is justified

or if I misunderstood their method.

2.

I do not understand the rationale behind the approximation by the

authors explained surrounding eq. (26). It is clear that a Gaussian

approximation for the stationary membrane potential distribution is

good as long as the mean input is sufficiently far from the threshold.

But I do not understand why they are using expressions (18) and (19)

directly to compute the mean and variance of the equilibrium

distribution by eq. (22), instead of using equations (24) and (25)

for k = 1 and k = 2. Unsing the latter, they would correctly

include the additional temporal filtering of the moments.

In their approach, the only temporal filtering seems to be coming

from the slow synaptic currents.

The approximation by the authors, instead, seems to be identical to

the Arrhenius approximation used in Brunel 2000, eq. 22 for the

instantaneous firing rate.

3.

I was missing a comparison of the methods used by the authors

to the direct simulation of the spiking network. At least for

some small examples that can still be simulated. Otherwise

it is hard to see how good the made approximations are, in

particular regarding my concerns about the correct treatment of

the time-dependent behavior, raised above.

Minor points:

1.

Eq. (26) is not correctly normalized. Factor 2 missing in the

prefactor.

2.

The maximum entropy approach should be explained more

explicitly in the methods part.

3.

Typos:

the index i has not limited to labeling single neuron

eq. (22) time dependence t on left side should be v

same in eq. (26)

**Have all data underlying the figures and results presented in the manuscript been provided?**

Reviewer #1: Yes

Reviewer #2: Yes

Reviewer #3: Yes

PLOS authors have the option to publish the peer review history of their article (what does this mean?). If published, this will include your full peer review and any attached files.

Reviewer #1: No

Reviewer #2: No

Reviewer #3: No

---

## [Decision Letter · Decision Letter 1]

29 Jan 2020

Dear Dr. Tao,

your revised manuscript has been already evaluated by the reviewers. They appreciated your effort in producing a very much improved resubmitted version. However, there are still some important issues outstanding. We would therefore like to ask you to modify the manuscript according to the review recommendations before we can consider your manuscript for acceptance. Your revisions should address the specific points made by each reviewer, and we encourage you to respond to particular issues raised.

We cannot make any decision about publication until we have seen the revised manuscript and your response to the reviewers' comments. Your revised manuscript is also likely to be sent to reviewers for further evaluation.

Sincerely,

Ernest Montbrió, Ph. D.

Guest Editor

PLOS Computational Biology

Lyle Graham

Deputy Editor

PLOS Computational Biology

Reviewer's Responses to Questions

**Comments to the Authors:**

Reviewer #1: It's fine

Reviewer #2: The authors have put considerable effort into clarification of their model, which is appreciated. However, they could still do a better job in places:

-As far as I’m able to see, they don’t model individual neurons in this paper: no details are given of such a simulation and in the response to reviewer 3 they refer to other work. I find it confusing to refer to a large-scale V1 Model (l 348), and then to proceed by giving equations for individual neurons that suggest that a large-scale model of individual neurons has been implemented. The authors even state a number of neurons (l 170), but there is no evidence that for the purpose of this paper such a model was implemented. I propose that the authors state very clearly upfront that any modelling here is done at the population level, and that the individual neuron models only serve as a step towards the final population model, and that when they throw around numbers like 10^6 that they relate to a number of neurons that a comparable model would comprise if it were built.

-I find it somewhat misleading to provide detailed descriptions of synaptic currents Eqs 2, 3, 6, 7, when it is clear from the Master equation that delta synapses are used (Eq. 15). The model clearly implements instantaneous transitions and does not contain any other dimension than the membrane potential. So, what is the point of a relatively lengthy section on individual neuron dynamics that is considerably simplified later on at the population level. For the purpose of this paper they could simply start off with delta synapses.

-I agree that rate-based models get fast transients wrong, and that may be a good argument for using them. I don’t see any evidence that this is the case in this particular model, which begs the question: why model an already complex network with complex methods when simpler ones suffice?

-I admire the thinking about the closure method, the idea behind it is much clearer now.

However, I’m in two minds about whether I personally would adopt the proposed solution here. A solution of the full system of coupled Fokker-Planck equations is not particularly time consuming, and not hard to implement. Implementing the moment closure method would need serious extra work on top of that, which overall may not save time.

On the other hand, an understanding of network dynamics in terms of a few dominant nodes is valuable, in particular when models like these will be part of larger networks. These considerations should not stand in the way of publication: the authors should make their case and take up will follow, or not.

-I also appreciate the authors’ ambition to create a larger, more complex network.

-I recommend that they publish their source code.

I think the appendices should not be in supporting material, but in the main text (as appendices). They are important.

Reviewer #3: The method is explained better in the revised manuscript,

thanks to the additional appendices.

The appendices are currently supplied as separate docx files,

which are hard to read and not displayed properly on non-windows

systems. Also, they will likely no be typeset by the

professional typesetters. I would like to encourage the authors

to include these appendices in the main manusctipt, so that

they will show as properly typeset as a pdfs.

Minor points:

In the appendix D there are some point unclear still.

1. The authors use two notations, \\rho_st and \\rho_eq.

I guess both should be identical.

2. In line 92 the authors state the form of the stationary

solution. This is specified in terms of an integral,

involving only the drift term, noted D^(1) here.

The authors do not state the form of D^(1), neither

refer to an equation where it is defined.

Also, it seems to me that the expression is incorrect.

What the authors likely mean is the method in Section 5.2

of Risken, "The Fokker-Planck equation"

rho_st(x) ~ e^\\Phi(x)

\\Phi(x) = \\int^x D^{1}(x') / D^{2}(x') dx'

where D^{1} is the drift and D^{2} the diffusion

coefficient.

What remains unclear in this appendix is whether the

authors use the stationary distribution with

standard boundary conditions or whether they use

the one that includes the absorbing boundary condition

at the threshold and the re-insertion of the flux

at reset. From the main text, it seems they use the

latter. This should be clarified in the Appendix

by referring back to section C of the Appendix.

3. It is unclear to me if the term "maximum entropy closure"

is indeed justified for the method the authors use.

This question arises, because the authors state in

Appendix C in line 69 "Furthermore, we induce an

equilibrium firing rate to solve the equilibrium

probability distribution."

It is not clear how they define the equilibrium

rate for a non-stationary Fockker-Planck equation.

More fundamental, my question is this:

Is the author's method identical to maximizing, at

each time point, the Shannon entropy of the voltage

distribution under the constraints that the moments

obey the constraints stated in line 114 of the Appendix?

If yes, this is not obvious to me from the given

material.

4. In line 101 of the Appendix, the authors call h(rho)

the density function. I think it would be helpful

to state here that this is the Kullback-Leibler divergence.

Typos:

period (.) missing after eq. 23

line 508

space too much in "point ."

line 592:

broken senstence:

"We call this closed-form by the augmented ODE system. found that in the simulations we performed in this paper, good performance can be achieved with only 4 augmented variables ..."

**Have all data underlying the figures and results presented in the manuscript been provided?**

Reviewer #1: None

Reviewer #2: Yes

Reviewer #3: Yes

PLOS authors have the option to publish the peer review history of their article (what does this mean?). If published, this will include your full peer review and any attached files.

Reviewer #1: No

Reviewer #2: No

Reviewer #3: No
---

## [Decision Letter · Decision Letter 2]

29 Apr 2020

Dear Dr. Tao

We are pleased to inform you that your manuscript 'Dimensional Reduction of Emergent Spatiotemporal Cortical Dynamics via a Maximum Entropy Moment Closure' has been provisionally accepted for publication in PLOS Computational Biology.

Best regards,

Ernest Montbrió, Ph. D.

Guest Editor

PLOS Computational Biology

Lyle Graham

Deputy Editor

PLOS Computational Biology

Reviewer's Responses to Questions

**Comments to the Authors:**

Reviewer #2: The authors have addressed the comments raised in earlier versions.

Reviewer #3: Thanks to the authors for clarifying the explanation of the

method. The additional material finally clarified the

method by the authors as replacing the time-dependent

solution for the probability by the one that, given

the time-dependent moments, is closest to the stationary

solution of the non-equilibrium problem.

I find the name "maximum-entropy method" irritating for this

method, because this name suggest that, along with the standard

argument by E.T. Jaynes 1957, the authors would maximize the

entropy of the distribution under the constraints that the

moments take the given values.

Instead, the authors minimize the Kulback-Leibler divergence

between the trial distribution and the stationary distribution

under the constraints that the moments take the values in the

non-stationary setting.

If the first two moments of the distribution are given,

the maximum entropy solution would obviously be a Gaussian

distribution if the standard form of Shannon's entropy had

been employed.

The form the authors use, instead, is eq. C.3, the stationary

solution of the LIF model, as derived e.g. by Amit & Brunel 1997.

The reason for the difference is obviously the presence

of the firing and reset rule (in absence of both, eq. C.3 would

reduce to a Gaussian).

More fundamentally, this is the stationary distribution of a system

that is not in thermodynamic equilbrium -- it violates detailed

balance.

So in contrast to the usual construction of the unique

maximum entropy solution, which has the meaning of a

distribution which makes no prior assumptions apart from

the known moments (see e.g. Jaynes 1957), this meaning

cannot (in the current form) be attributed to the presented

work; the latter rather seems to me like an ad-hoc procedure

so far.

It may well be that I am missing some point here.

I do not want to delay the publication of the manuscript

further -- even though the formal justification of the method

right now is not understandable to me, there may still be use

of the method; time will show.

Still, I would like to suggest to the authors to clearly explain

the difference between what they call "Maximum entropy procedure"

and what is commonly understood under this term, say, following

the view of the subjectivist view on therodynamics (Jaynes et al 1957).

Alternatvely, tracing back their procedure to maximizing the

Shannon-entropy under the given constraints would be of course

even more welcome.

typos:

l. 1075 space missing before word "changing"

l. 1078 brackets mssing around argument of first exp function

**Have all data underlying the figures and results presented in the manuscript been provided?**

Reviewer #2: Yes

Reviewer #3: Yes

PLOS authors have the option to publish the peer review history of their article (what does this mean?). If published, this will include your full peer review and any attached files.

Reviewer #2: No

Reviewer #3: No

---

## [Editor Report · Acceptance letter]

26 May 2020

PCOMPBIOL-D-19-01160R2 

Dimensional Reduction of Emergent Spatiotemporal Cortical Dynamics via a Maximum Entropy Moment Closure

Dear Dr Tao,

I am pleased to inform you that your manuscript has been formally accepted for publication in PLOS Computational Biology. Your manuscript is now with our production department and you will be notified of the publication date in due course.

With kind regards,

Laura Mallard
